# Robust quantum valley Hall effect for vortices in an interacting bosonic quantum fluid

O. Bleu[1], G. Malpuech[1] & D. D. Solnyshkov[1]

Topologically protected pseudospin transport, analogous to the quantum spin Hall effect, cannot be strictly implemented for photons and in general bosons because of the lack of symmetry-protected pseudospins. Here we show that the required protection can be provided by the real-space topological excitation of an interacting quantum fluid: a quantum vortex. We consider a Bose-Einstein condensate at the $\Gamma$ point of the Brillouin zone of a quantum valley Hall system based on two staggered honeycomb lattices. We demonstrate the existence of a coupling between the vortex winding and the valley of the bulk Bloch band. This leads to chiral vortex propagation on each side of the zigzag interface between two regions of inverted staggering. The topological protection provided by the vortex winding prevents valley pseudospin mixing and resonant backscattering, allowing a truly topologically protected valley pseudospin transport.

[1] Institut Pascal, PHOTON-N2, University Clermont Auvergne, CNRS, 4 Avenue Blaise Pascal, 63178 Aubière Cedex, France. Correspondence and requests for materials should be addressed to D.D.S. (email: dmitry.solnyshkov@uca.fr)

Topological defects are a distinctive feature of quantum fluids[1]. Such real space excitations are stable and cannot be removed by a continuous transformation, which is called topological protection. They are known for more than fifty years and determine the fluid properties, for example, in the Berezinskii–Kosterlitz–Thouless phase transition in Bose-Einstein Condensates (BECs)[2].

Since the eighties, the concept of topology has been applied to reciprocal space. The topology of Landau levels[3,4] and more generally of Bloch bands[5] has been shown to determine the spectacular properties of topological insulators. In this case, the single-particle energy bands of the system are described by topological invariants[4] (such as the Chern number). The field expanded even further with the discovery of the quantum spin Hall effect and of the associated class of $\mathbb{Z}_2$ topological insulators[6]. Indeed, if one considers spinor particles in a lattice (electrons for instance), the parity of the Chern number computed using only one spin component is a topological invariant for a Hamiltonian verifying Time-Reversal Symmetry (TRS)[7].In that case the bulk-boundary correspondence applies and guarantees on the interface with a trivial insulator the presence of a pair of counter-propagating spin-polarized states, which because of TRS do not couple with each other.

This triumph of topology was followed by the attempts to extend the concept of topologically protected spin transport to other types of two-level systems which can be mapped to a pseudospin representing either an internal degree of freedom (photon polarization) or an external one (angular momentum, valley[8], etc.). However, for photons, TRS acts differently from fermions[9] and rigorously, there is no symmetry-protected photonic quantum spin Hall effect. This can be clearly visualized by explicitly considering the photonic spin-orbit coupling due to the energy splitting between TE (transverse-electric) and TM (transverse-magnetic) modes[10,11]. It respects TRS, but it has a double winding which couples counter-propagating spin-polarized photonic modes. The realization of topological spin transport for light therefore requires to fabricate a structure where the TE-TM splitting is weak, which is possible but very demanding[12,13]. Other degrees of freedom, like the angular momentum of photons in lattices of ring cavities have been considered[14,15], with the formal problem that no specific symmetry protects this pseudospin from disorder. Finally, the quantum valley Hall (QVH) effect in staggered honeycomb lattices uses the valley pseudospin[8,16,17] to which one can associate valley Chern numbers[18]. QVH has been evidenced experimentally in electronic systems[19] and recently considered in a large series of works in topological photonics[20–25]. In these systems the dissipation mechanism is the inter-valley scattering[26]. Even if it is argued to be weak, any defect localized on the interface induces a back-scattering of the edge modes[27] (see also Supplementary Movie 1). It formally limits the meaning of the valley Chern number as a topological index, and leads to the Anderson localization of the 1D edge states.

Recently, the topology of quantum fluids in real space and of the band in the reciprocal space have been fruitfully combined in the field of topological superconductors and superfluids[6,28].The collective excitations are split off by the superconducting gap, which is topologically non-trivial for specific pairing, creating topological edge states. A vortex necessarily contains such edge states, which can be Majorana fermions[29] protected by the particle-hole symmetry. Many other solitonic[30–35] and vortex[36] solutions were found in non-trivial topologies, but for BEC systems, the chiral behavior has been mostly discussed for weak Bogoliubov excitations[37–42].

In this work, we propose an original combination of real and reciprocal space topologies, creating a truly protected pseudospin current in a bosonic system. Here, the topological phase and the edge pseudospin currents are not protected by a symmetry of the Hamiltonian, but by the winding number of the quantum vortices (real space topological quantum number[43]). We consider a BEC at the Γ point of the Brillouin zone of a QVH system based on two staggered honeycomb lattices. We demonstrate the existence of a coupling between the vortex winding and the valley of the bulk Bloch band. This coupling leads to chiral vortex propagation on each side of an interface between two regions with inverted staggering, with a true topological protection against back-scattering, contrary to the interface states of the non-interacting Hamiltonian. This configuration can be seen as a quantum spin Hall effect analog, but where the role of spin is played by the winding of the vortices. Our results apply to polariton condensates in recently fabricated polariton honeycomb lattices[44] and to atomic BECs in optical lattices[45].

## Results

**Non-interacting QVH**. We consider an interface between two honeycomb lattices with opposite staggering, each being described by a tight-binding (TB) Hamiltonian:

$$H_k = \begin{pmatrix} \Delta & -Jf_{\mathbf{k}} \\ -Jf_{\mathbf{k}}^* & -\Delta \end{pmatrix}, f_{\mathrm{k}} = \sum_{j=1}^{3} \exp\left(-i\mathbf{k}\mathbf{d}_{\phi_j}\right) \qquad (1)$$

where $\mathbf{k}$ is the wave vector, $\mathbf{d}_{\phi_j}$ is the vector connecting nearest neighbor sites, $2\Delta = E_B - E_A$ is the energy difference between A and B sites, and $J$ is the tunneling coefficient. A non-zero $\Delta$ opens a bandgap and implies opposite Berry curvatures in $K$ and $K'$ valleys. If the gap is sufficiently small, the Berry curvature is localized in each valley giving valley Chern numbers: $C_{K,K'} = \pm 1/2$. The number of chiral states in each valley at the zigzag interface is defined by the domain wall topological invariant:[46] $N_{K,K'} = C_{K,K'}(l) - C_{K,K'}(r) = \pm 1$ (where $l$ and $r$ stand for the left and right domains). This results in the presence of one chiral state in each valley, with opposite group velocities (QVH effect). However, these valley states, degenerate in energy, are not symmetry-protected, which means that the backscattering due to valley mixing by disorder is not forbidden for single particles.

**Quantum vortices**. The BEC can be described by a single-particle wavefunction (WF) $\psi$ (the order parameter). In the mean-field approximation, $\psi$ is the solution of the Gross-Pitaevskii equation (GPE), including interparticle interactions:

$$i\hbar \frac{\partial \psi}{\partial t} = -\frac{\hbar^2}{2m}\Delta\psi + \alpha|\psi|^2\psi + U\psi - \mu\psi \qquad (2)$$

where $m$ is the particle mass, $\alpha$ is the interaction constant, $U$ is the external potential, and $\mu$ is the chemical potential. The existence of $\psi$ imposes the irrotationality of this bosonic quantum fluid: $\nabla \times \mathbf{v} = 0$ everywhere, except zero-density points. The condensate velocity is given by $\mathbf{v} = \hbar\nabla\varphi/m$ ($\varphi = \arg\psi$). The phase winding around the zero-density points where $\psi = 0$ is fixed by the single-valuedness of $\psi$: $\oint \nabla\varphi dl = 2\pi p$, where $p$ is the winding number. The solutions with non-zero $p$ are called vortices, and their characteristic size is determined by the healing length $\xi = \hbar/\sqrt{2\alpha nm}$. We shall consider single-winding vortices ($p = \pm 1$) in staggered honeycomb lattices. Vortices with higher winding are energetically unstable and split into single-winding vortices[2].

**Winding-valley coupling**. First, we shall demonstrate that the core of a vortex with a given winding corresponds to a certain valley ($K$ or $K'$) of the single-particle dispersion of staggered

graphene, that is, the existence of winding-valley coupling for vortices.

Let us consider the core of a sufficiently large vortex ($\xi \gg a$, where $a$ is the distance between nearest neighbors), where the density is necessarily small and the interactions can be neglected. To minimize the on-site energy given by $E = E_A|\psi_A|^2 + E_B|\psi_B|^2$, the WF is mostly localized on the sites of the A type, which have lower energy (assuming $E_A < E_B$). In the limit of a large gap, $\Delta \gg J$, only the A-sites are populated and therefore the Bloch function in the vicinity of the vortex center is $(1, 0)^T$. We can obtain the corresponding plane wave by Fourier transform of the WF $\tilde{\psi}(\mathbf{k})$ analytically (see Supplementary Note 1 for details). We find that the maximum value of the WF is achieved for $\mathbf{k} = K$ and $\mathbf{k} = K'$, depending on the vortex winding $p$. Thus, both the Bloch wave and the plane wave part of the WF in the core of a vortex of a given winding define a state corresponding to a certain well-defined valley of the single-particle dispersion. The winding-valley coupling reads:

$$\tau = ps \qquad (3)$$

where $\tau = \pm 1$ is the valley number and $s = \mathrm{sgn}(\Delta)$ is the lattice staggering. This result is linked with the well-known optical selection rules in Transitional Metal Dichalcogenides[47] where the phase pattern at the $K$ point exhibits an angular momentum for each unit cell, determining the angular momentum of photons for a given valley. We note that if the vortex size becomes comparable with the lattice parameter $\xi \approx a$, the winding-valley coupling becomes reduced: the vortex acquires a fraction of a different valley. At the same time, it becomes localized on the hexagon center and its mobility is reduced.

To confirm our analytical solution, we have performed numerical simulations by solving the GPE beyond the TB approximation, with an explicit honeycomb lattice potential $U(\mathbf{r})$. Without losing generality, we consider all parameters as in ref. [27] (typical for exciton-polaritons[44]), but considering a quasi-conservative case. The neighbor distance is $d = 2.5\,\mu\mathrm{m}$, pillar radius $r = 1.5\,\mu\mathrm{m}$, $m = 5 \times 10^{-5}m_0$ ($m_0$ is the free electron mass), corresponding to $J \approx 0.25\,\mathrm{meV}$, and $\alpha n = 0.3\,\mathrm{meV}$. To find the WF of the vortex, we have introduced a relaxation term[48], preserving zeros of the WF ($\Lambda = 0.03$) (see Methods). The results of these calculations are shown in Fig. 1, where the black lines show the contours of the pillars corresponding to the potential $U(\mathbf{r})$ (panels (a)–(c)), and the white dashed line shows the 1st Brillouin zone (panels (d), (e)). To get the information on the vortex core, we apply spatial filtering using a Gaussian of size $w$. For large $w$, the image in the reciprocal space (Fig. 1(d)) is dominated by the condensate centered at the ground state ($\Gamma$ point). The ground state itself is empty, because the vortex imposes $v \neq 0$ everywhere. For smaller $w$ (Fig. 1(e, f)), the core of the vortex is centered at the $K$ points of the reciprocal space, while the $K'$ valleys are empty. Opposite results are obtained for opposite winding, confirming the valley-winding coupling for vortices.

**Vortex at the interface**. We have shown that the vortex WF in the reciprocal space has two contributions. Most of the condensed particles, far from the vortex core, are concentrated around the $\Gamma$ point (small $k$). These particles are practically unaffected neither by the presence of the lattice, nor by any possible interfaces. On the other hand, the core of the vortex is at the $K$ point, and we can expect interesting effects linked with the interfaces, where in the linear regime the states from the bulk $K$ points give rise to chiral propagative interface states (QVH states).

We calculate analytically the energy of the vortex as a function of both the wavevector of the core (dispersion) and of its position in real space, using the TB approximation and the grand canonical expression:[2]

$$E_v = \int \left( \frac{\hbar^2}{2m}|\nabla\psi|^2 + \frac{\alpha}{2}\left(|\psi|^2 - n\right)^2 \right)\mathbf{dR} \qquad (4)$$

Qualitatively, this expression is the difference between the energy of a system with a vortex and the energy of a system without a vortex (but with a condensate in the ground state with the unperturbed density $n$). The first step is to split the integral into 2 regions: the core ($|\mathbf{R}| \leq \xi$) and the outside zone ($|\mathbf{R}| > \xi$). In the second region, $|\psi|^2 \approx n$, and the only contribution to the vortex energy comes from the kinetic energy term, which gives the well-known logarithmic expression $E_v^{r>\xi} = \pi n\hbar^2\ln(1.46R_0/\xi)/m$ ($R_0$ is the system size).

In the vortex core, the presence of the lattice has to be taken into account. As we have shown above both analytically and numerically, the core of the vortex is a wavepacket centered at $k_0$ close to either $K$ or $K'$ (we take a Gaussian wavepacket $\psi_G$). We calculate its energy versus $k_0$ using the TB dispersion $E(\mathbf{k})$. The $X$ spatial direction, perpendicular to the interface, has to be treated in the real space ($x_0$ is the vortex center). The contribution to the kinetic energy is calculated as: $E_v^{\mathrm{kin},r<\xi}(x_0, k_0) = \int_{x_0-\xi}^{x_0+\xi}dx\int dk\psi_G^*\psi_0^*\hat{H}\psi_0\psi_G$, where $\psi_0(x, k_y)$ are the single-particle eigenstates of the lattice. These eigenstates are quantized in the $X$ direction. Their spatial overlap with the vortex core plays an important role. For the delocalized bulk states the overlap tends to zero with increase of the stripe width. On the other hand, the state localized at the interface (width $\kappa$) has a non-vanishing overlap and the contribution of this state dominates the dispersion of the vortex core. An example of such dispersion in the vicinity of the $K$ and $K'$ points is shown in Fig. 2 (a,b): the dispersion of the core (blue line) inherits the dispersion of the linear eigenstates at the interface (red dots), and therefore their valley-dependent propagation direction (chirality), as compared with the non-propagating bulk states with zero group velocity exactly at $K$ or $K'$ (black points).

The kinetic energy of the core also depends on the position of its center $x_0$: if the core is perfectly centered at the interface, the energy at $k_0 = K$ is exactly the same as that of the interface state. On the other hand, if the core is located in the bulk, its energy is that of the top of the valence band, determined by the energy splitting $E^{\mathrm{kin}}(x_0, k_0) = -\Delta$. The interface therefore represents a barrier of a height of the order of the gap $\Delta$, if only the kinetic energy is taken into account.

The contribution of the interactions to the vortex energy comes from the sensitivity of the vortex to the local changes of the density in the condensate. In the vortex core, the density $|\psi|^2$ is small as compared with the background density $n(\mathbf{r})$, and the integral reads: $E_v^{\mathrm{int},r<\xi} = \int_0^\xi \alpha n^2 2\pi r dr$. Thus, the vortices are attracted to lower-density regions minimizing the total energy of the system. The density of the condensate without a vortex depends on the local potential, which affects the density of the condensate at the scale given by the healing length $\xi$. Considering the interface as a Delta barrier $V_0\delta(x)$, the density of the condensate in its presence can be found as: [49] $n(x) = n_0(1 - \cosh^{-2}((x_c + |x|)/\xi'))$, where $x_c$ and $\xi'$ depend on $V_0$. The interaction energy of the vortex core as a function of $x_0$ therefore exhibits a minimum of width $\xi' \approx \xi$.

The sum of kinetic and interaction energies depends on the parameters of the system. Two examples of such dependence as a

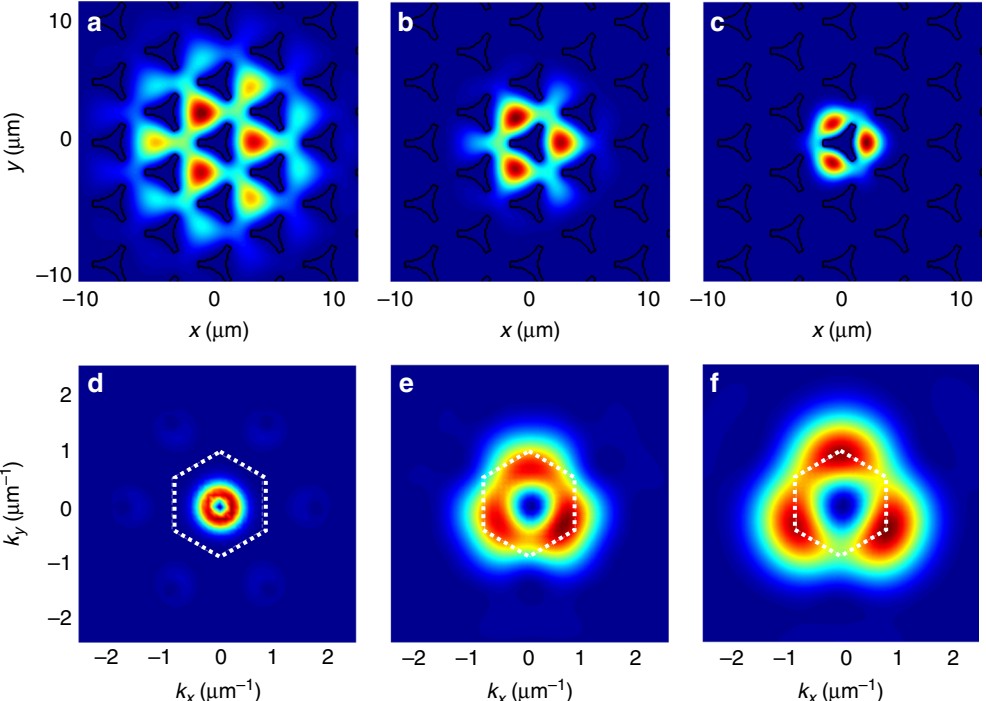

**Fig. 1** Valley-Vortex winding coupling. Numerical density profile of the vortex stationary solution in real (**a–c**) and reciprocal (**d–f**) space for different filtering scales ($w = 7, 3, 1\,\mu m$, respectively)

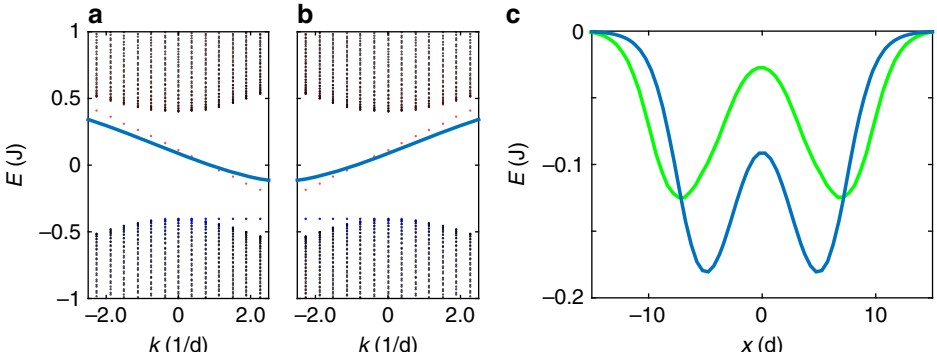

**Fig. 2** Valley polarized topological interface modes. **a**, **b** Energy of the vortex core at the interface as a function of its central wavevector, exhibiting valley chirality (Solid blue line; **a**-$K$, **b**-$K'$). Linear state dispersion on the interface (blue dashed) and in the bulk (black dots). **c** Energy of the vortex as a function of position. Blue and green curves correspond to a direct interface or a larger one constituted of 4 unstaggered zigzag chains centered at $x = 0$, respectively

function of $x_0$ are shown in Fig. 2(c) for $\xi > \kappa$. The vortex can be localized on either side of the interface, the latter acting as a barrier preventing the vortex from changing domain and valley (the only valley scattering mechanism remaining for vortices unless the winding-valley coupling is suppressed). Tunneling through the interface can occur through quantum-mechanical[50,51] or thermal[52] mechanisms: $P_{QM} \sim \exp(-nl^2)$ and $P_T \sim \exp(-\Delta/\sigma)$ ($l$—interface width, $\sigma$—broadening). These tunnelings are extremely small. It is moreover possible to increase the barrier size by inserting several zigzag chains with $\Delta = 0$ between the two staggered lattices, thus reducing the tunneling exponentially and making it negligible and increasing the robustness against point-like defects (see Supplementary Note 3 for details). The green line in Fig. 2(c) shows the vortex energy for a wide interface (4 zigzag chains). Vortex tunneling is thus restricted by the same condition as the observation of the edge states: $\sigma < \Delta$—the broadening of all sources should be smaller than the gap.

## Discussion

Our analytical results are fully confirmed by numerical simulations of vortex propagation along the interface using Eq. (2) (no $\Lambda$). The snapshots of one of such simulations are shown in Fig. 3 (see Supplementary Movie 2). We see that the vortex remains attached to the interface and propagates along, without being scattered backwards on the corners. An additional defect of 1 meV ($\sim 4J \gg \Delta$) and 1 μm width has been added on an interface pillar for comparison with the linear case, where it leads to strong backscattering[27]. This allows us to check that the vortex is indeed immune to backscattering thanks to the additional topological protection provided by its winding via the winding-valley coupling. A detailed analysis of the impact of the defect size and interface thickness on the scattering processes is shown in Supplementary Note 4 (see also Supplementary Movies 3-6). We stress, however, that in contrast with electronic quantum spin Hall insulators, where the particle number is conserved, counter-propagating vortices can annihilate.

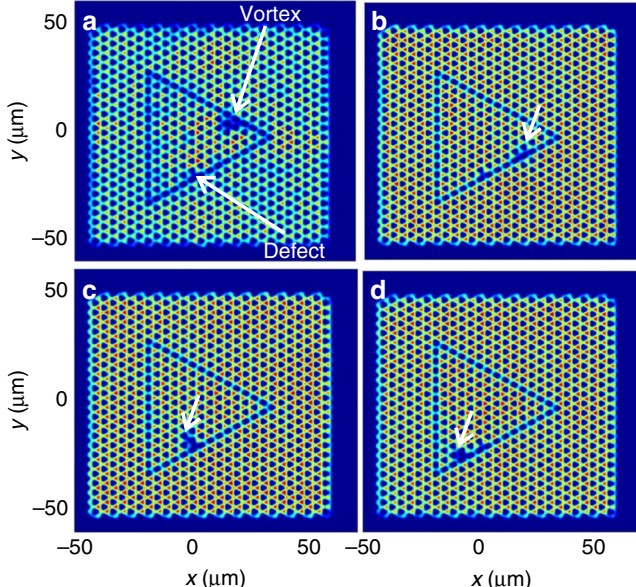

**Fig. 3** Robust chiral motion of a quantum vortex on the interface. Snapshots of the vortex propagation along the interface, showing the spatial density distribution $|\psi(x, y)|^2$. Panels (**a–d**) correspond to instants $t =$ 0, 210, 390, 480 ps

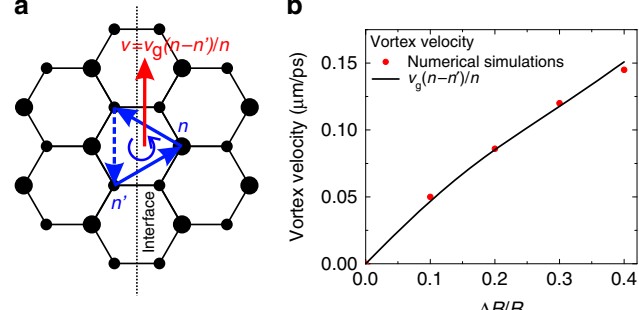

**Fig. 4** Quantum vortex velocity. **a** A vortex at an interface and its net velocity. **b** Vortex velocity as a function of the gap size. Red points—numerical results, black—analytical solution

The group velocity of the interface states is $v_g = \hbar^{-1}\partial E/\partial k = 0.7 \times 10^6\ \mathrm{ms}^{-1}$. This is the velocity with which the WPs at the interface can be expected to propagate in this particular lattice. Interestingly, the vortex velocity is different from $v_g$. We stress that it is also different from what can be calculated for the vortex rolling effect (see Supplementary Note 2). Indeed, in our calculation we were assuming that only one type of the sites is occupied for a given staggering. However, as shown in a scheme in Fig. 4(a), the interface represents a violation of a perfect staggering, and thus the higher-energy sublattice acquires a density estimated as $n' = 2n/\left(1 + \left(\Delta + \sqrt{\Delta^2 + 4J^2}\right)^2/4J^2\right)$. The resulting velocity, reduced with respect to that of the linear interface states, is given by:

$$v = v_g(n - n')/n \tag{5}$$

We plot the dependence of $v$ on the pillar size ratio $\Delta R/R$ (determining the gap size $\Delta$) in Fig. 4(b). Red dots show the results of numerical simulations. Black line is the analytical solution given by Eq. (5), where $v_g$ and $\Delta$ are taken from numerical simulations in linear regime. We see that it corresponds almost perfectly to the points (exact numerical solution) while there are no fitting parameters. This confirms the validity of our interpretation. For a smooth interface, $n' \approx n$ and the velocity is reduced even more (see Supplementary Note 2).

Single-domain configurations with QVH edge states have been considered for single particles in the past[53]. While the edge states provide a similar dependence $E_v(k_0)$ for vortices (giving rise to associated velocity along the edge), such configuration does not exhibit the same localization potential in transverse direction $E_v(x_0)$ as that shown in Fig. 2(c), and the vortex either crosses the interface and disappears or enters the bulk (see Supplementary Note 4, Supplementary Movies 7,8). Therefore, the abrupt domain wall between two opposite-staggered lattices that we consider is really an optimal configuration for vortices.

The mean-field approximation we are using neglects quantum and thermal fluctuations, which reduce the coherence length of the condensate. For the particular system of exciton-polaritons in

GaAs cavities at 5 K, the effect of thermal fluctuations is reduced because of the strong decoupling from the phonon reservoir[54]. The quantum fluctuations give a theoretical limit for the coherence length of the order of 1 mm[55], which makes the mean-field GPE a good approximation at the scale of the lattice we consider (100 μm). The main sources of broadening, limiting the possibilities of experimental observation, are therefore the disorder and the finite lifetime. If the broadening becomes so strong that it closes the gap, the winding-valley coupling is suppressed. This can be considered as a valley scattering mechanism for a vortex, corresponding to the same restriction for chiral vortex propagation as already discussed above: $\sigma < \Delta$. On the other hand, the winding inversion for the vortex is suppressed even stronger than its tunneling across the barrier, that is, by a factor $\exp(-N)$, where $N \sim 10^4$ is the total number of particles in the condensate. Such mechanism is therefore completely improbable and does not add any restrictions to the experimental conditions.

To conclude, our work addresses the behavior of quantum fluids in topologically non-trivial systems. It highlights a new combination of topological effects: real space topological effect characterized by vortex winding number and momentum space topology characterized by the valley Chern number. We see that the properties of the single-particle dispersion of the interface states are inherited by the vortex solution of the non-linear equation via the core, the vortex providing protection against backscattering by localized disorder on the interface. We demonstrate that this combination allows to achieve topologically robust QVH effect. These results are promising for the development of a new field of vortextronics, where the information will be carried by vortices. The possibility to create chiral pathways for vortices and to automatically sort them according to their winding is crucial for information treatment.

## Methods

**Numerical simulations.** We used third-order Adams-Bashforth method for the time integration of the Gross-Pitaevskii Eq. (2), both with the relaxation term to find the stationary vortex solution, and without the relaxation term to study the vortex behavior. The Laplacian term was replaced by a double Fourier transform, in order to obtain an efficient parallelization on the Graphics Processing Unit (using nVidia Compute Unified Device Architecture—CUDA).

To find the stationary vortex wavefunction in presence of a honeycomb lattice potential, we solve the damped Gross-Pitaevskii equation: [48]

$$i\hbar\frac{\partial\psi}{\partial t} = (1 - i\Lambda)\left(-\frac{\hbar^2}{2m}\Delta\psi + \alpha|\psi|^2\psi + U\psi - \mu\psi\right) \tag{6}$$

where $\Lambda = 0.03$ is the dimensionless damping coefficient. This equation guarantees that a stationary solution with an energy $\mu$ persists, whereas any perturbations to this solution with higher energies decay, with the characteristic decay rate proportional to the energy deviation $\Gamma = \langle H \rangle - \mu$. We start with a wavefunction $\psi \sim \tanh(r/r_0)\exp(ip\varphi)$, where $\varphi$ is the polar angle, $p = \pm 1$ and $r_0$ is of the order of

expected healing length $\xi$ (several micrometres). The damped equation conserves the zeroes of the wavefunction, because in the point $r = 0$ where $\psi = 0$ the right part of the equation vanishes and thus $\partial \psi / \partial t = 0$. The wavefunction cannot therefore evolve towards the ground state, so it stabilizes at a stationary solution with winding $p = \pm 1$.

The energy of the vortex (Fig. 2) was calculated from a tight-binding model for a graphene ribbon with an interface between two opposite staggerings.

**Code availability**. The code used for numerical simulations based on the Gross–Pitaevskii equation is available from the corresponding author upon reasonable request.

## Data availability

The data generated with the above code are available from the corresponding author upon reasonable request.

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

## Acknowledgements

We acknowledge the support of the ANR project "Quantum Fluids of Light" (ANR-16-CE30-0021), of the ANR Labex Ganex (ANR-11-LABX-0014), and of the ANR program "Investissements d'Avenir" through the IDEX-ISITE initiative 16-IDEX-0001 (CAP 20-25). D.D.S. acknowledges the support of IUF (Institut Universitaire de France).

## Author contributions

O.B. performed the analytical and tight-binding calculations. G.M. supervised the project. D.D.S. did the numerical simulations and supervised the project. All authors contributed to achieve the main results and to the manuscript.

## Additional information

**Competing interests:** The authors declare no competing interests.

