## [Peer Review File · Nature Communications]

Reviewers' comments:

Reviewer #1 (Remarks to the Author):

By using the quantum valley Hall (QVH) background, the authors propose to realize the vortex version of QVH effect in an interacting bosonic quantum fluid. As a specific example, a valley-winding coupling zigzag domain wall has been showcased.

The results are solid, the research is timely, and the discovery is exciting. Therefore, I strongly recommend the publication of this manuscript in Nature Communications. Before publication, here are some suggestions that I encourage the authors to consider.

1. The authors seem to claim that the QVH effect of vortices is more robust than that of electrons/photons. However, I think they are similarly robust. Say, if the lattice-scale effect is sufficiently strong, the QVH edge or domain-wall bands (kinetic energy) would disappear (valleys are not well defined any more), and thus the vortex-valley coupling would be destroyed. Since the paper is really a QVH effect of vortices, why not using "QVH" instead of "QSH analogy" in the title?

2. Related to the above question, how is the effect at the edge of a single-domain sample, where the inter-valley coupling is strong?

3. Refs. 8 and 16 are cited for QVH states. However, ref. 8 is on the geometric valley Hall effect (in which the integration of valence-band Berry curvature is not taken into account). To the best of my knowledge, PRL 106, 156801 (2011) is the first proposed the term QVH. Additionally, Nature Physics 14, 111–113 (2018) has discussed quite a few "particles/waves" in which their QVH effects have been observed. It would be interesting to compare them.

4. What about if the winding number becomes 2, 3, 4, ...

Reviewer #2 (Remarks to the Author):

The manuscript by Bleu et al. investigated the topological excitation of an interacting quantum fluid subject to a staggered honeycomb lattice potential with topological interface. It is known that the single particle spectrum in such potential features valley helical mode in the bulk bandgap, which, however, are not protected from backscattering by any symmetry. In the interacting quantum fluid, the authors found that while most of the condensed particles are concentrated around the Gamma point, not much unaffected by the lattice and topological interface, the vortices are located at the K and -K points. The authors discovered a coupling between the vortex winding and the valley index, such that the vortices at the two valleys have opposite winding, a factor that protects the vortices from backscattering. They further showed that the vortex can be energetically trapped by the topological interface, where numerical simulations demonstrate robust vortex propagation against defect. This is a very novel piece of work, which can be readily explored in polariton honeycomb lattices and atomic Bose-Einstein condensates in optical honeycomb lattices. With the following comments properly addressed, we would like to recommend publication of the manuscript in Nature Communications.

1) It would be interesting to examine how the conclusions change when the sharp topological interface in the staggered honeycomb lattice is replaced by a smooth one (i.e. with Δ varying smoothly from

positive on one side to negative on the other). What properties of the vortices can be affected by the width of the topological interface?

2) Concerning the single particle behaviours, the staggered honeycomb lattice with a sharp topological interface used by the authors was shown to be equivalent to an open ended staggered honeycomb lattice subject to an edge potential with strength equal to the hopping amplitude (c.f. Phys. Rev. Lett. 102, 096801 (2009)). When the quantum fluid is subject to the latter potential, will there be vortices trapped at the open edge as well with the winding-valley coupling?

3) On page 6, the authors address the winding-valley coupling by taking into account only the 3 atoms of the A type of the central hexagon (in supplementary material). When the vortex core size is large compared with lattice constant, will it be necessary to take into account more atoms, e.g., next nearest neighbour atoms? How does the vortex core size affect the coupling quantitatively?

4) In Fig. 1, does the vortex core lie exactly at an A type lattice site? It will be helpful to show the lattice sites together with the density profile. And why the vortex density profile has no C3 symmetry?

5) The authors should explicitly explain why and how they filter the vortex core using Gaussian of different size w in Fig. 1a-c.

6) In the last paragraph of page 9, what does "kappa" mean in the second line? It seems that there is no definition given in the main text.

7) It would be helpful to have some discussions on the finite temperature effect.

Reviewer #1 (Remarks to the Author):

By using the quantum valley Hall (QVH) background, the authors propose to realize the vortex version of QVH effect in an interacting bosonic quantum fluid. As a specific example, a valley-winding coupling zigzag domain wall has been showcased.

The results are solid, the research is timely, and the discovery is exciting. Therefore, I strongly recommend the publication of this manuscript in Nature Communications. Before publication, here are some suggestions that I encourage the authors to consider.

We thank the referee for the attentive reading of our manuscript, for its positive appreciation, and for the constructive proposals.

1. The authors seem to claim that the QVH effect of vortices is more robust than that of electrons/photons. However, I think they are similarly robust. Say, if the lattice-scale effect is sufficiently strong, the QVH edge or domain-wall bands (kinetic energy) would disappear (valleys are not well defined any more), and thus the vortex-valley coupling would be destroyed. Since the paper is really a QVH effect of vortices, why not using "QVH" instead of "QSH analogy" in the title?

The referee is right in the statement that both effects disappear if the valleys are not defined any more. However, we would like to stress that smaller perturbations, like localized defects, which are naturally present in any structure, couple the valleys without destroying them. This coupling induces a backscattering in the QVH effect with electrons or photons, but does not lead to the backscattering of the vortices, which are protected by the winding-valley coupling. This is one of the main results of our work.

As to the analogy with QSH, the precise sense of the analogy is stated in the manuscript as follows:

This configuration can be seen as a quantum spin Hall effect analog, but where the role of spin is played by the winding of the vortices.

The spin of the electrons provides the protection against the scattering on non-magnetic defects in QSH effect. In our case, it is the winding of the vortices which provides a similar protection.

For the above reasons, we find it important to keep the present title, which provides a correct interpretation of our results.

2. Related to the above question, how is the effect at the edge of a single-domain sample, where the inter-valley coupling is strong?

We thank the referee for this inspiring question, which is linked with the question 2 of the second referee (see below). We have carried out additional simulations to elucidate this problem, and the results have been included in the Supplementary. They are also commented in the main text.

At the edge of a single-domain sample the coupling of the valleys leads to the formation of non-dispersive edge states. Therefore, one cannot expect a chiral vortex propagation without additional changes. This is indeed confirmed by simulations. The case of such sample with an additional potential is discussed below (see the reply to Referee 2).

3. Refs. 8 and 16 are cited for QVH states. However, ref. 8 is on the geometric valley Hall effect (in which the integration of valence-band Berry curvature is not taken into account). To the best of my knowledge, PRL 106, 156801 (2011) is the first proposed the term QVH. Additionally, Nature Physics 14, 111–113 (2018) has discussed quite a few "particles/waves" in which their QVH effects have been observed. It would be interesting to compare them.

We thank the referee for these references, which we now cite in appropriate places of the main text, within the discussion of the history of quantum valley Hall effect and its limitations:

Finally, the quantum valley Hall (QVH) effect in staggered honeycomb lattices uses the valley pseudospin $\text{\textcolor{red}\cite{Niu2007,Macdonald2011,ren2016topological}}$ to which one can associate valley Chern numbers $\text{\textcolor{red}\cite{zhang2013valley}}$.

In these systems the dissipation mechanism is the inter-valley scattering $\text{\textcolor{red}\cite{Zhang2018}}$.

4. What about if the winding number becomes 2, 3, 4, ...

This is a very interesting question. While we could study the dispersion of such wavepacket theoretically in the linear regime (Gauss-Laguerre wavepacket at the interface), in the nonlinear regime the high-winding vortices are unstable. They separate into several single-winding vortices. We now comment on this in the main text:

Vortices with higher winding are energetically unstable and split into single-winding vortices $\text{\textcolor{red}\cite{Pitaevskii}}$.

Reviewer #2 (Remarks to the Author):

The manuscript by Bleu et al. investigated the topological excitation of an interacting quantum fluid subject to a staggered honeycomb lattice potential with topological interface. It is known that the single particle spectrum in such potential features valley helical mode in the bulk bandgap, which, however, are not protected from backscattering by any symmetry. In the interacting quantum fluid, the authors found that while most of the condensed particles are concentrated around the Gamma point, not much unaffected by the lattice and topological interface, the vortices are located at the K and -K points. The authors discovered a coupling between the vortex winding and the valley index, such that the vortices at the two valleys have opposite winding, a factor that protects the vortices from backscattering. They further showed that the vortex can be energetically trapped by the topological interface, where numerical simulations demonstrate robust vortex propagation against defect.

This is a very novel piece of work, which can be readily explored in polariton honeycomb lattices and atomic Bose-Einstein condensates in optical honeycomb lattices. With the following comments properly addressed, we would like to recommend publication of the manuscript in Nature Communications.

We thank the referee for the very careful reading of our manuscript and for stimulating questions, to which we reply below:

1) It would be interesting to examine how the conclusions change when the sharp topological interface in the staggered honeycomb lattice is replaced by a smooth one (i.e. with Δ varying smoothly from positive on one side to negative on the other). What properties of the vortices can be affected by the width of the topological interface?

We have made the corresponding simulations which are now discussed in the Supplementary and commented in the main text.

Qualitatively, the vortices are slowed down according to Eq. (5), because the densities on A and B sites become almost equal for a smooth interface. Moreover, such interface provides a reduced topological protection, which is ensured by the size of the gap. We comment on this in the main text:

For a smooth interface, $n \approx n'$ and the velocity is reduced even more (see Supplementary Note 3).

And in the Supplementary:

For a smooth interface, the difference between n and n' is strongly reduced and the vortex velocity drops down, limiting the possibilities of experimental observation and practical applications. The vortex also becomes less bound to the interface and its propagation can be easily perturbed by defects, because the topological protection, provided by the gap size, is reduced. The abrupt transition between the staggering domains therefore seems to be a better configuration.

2) Concerning the single particle behaviours, the staggered honeycomb lattice with a sharp topological interface used by the authors was shown to be equivalent to an open ended staggered honeycomb lattice subject to an edge potential with strength equal to the hopping amplitude (c.f. Phys. Rev. Lett. 102, 096801 (2009)). When the quantum fluid is subject to the latter potential, will there be vortices trapped at the open edge as well with the winding-valley coupling?

We thank the referee for this interesting question which is linked with the question 2 of referee #1. We have made additional simulations which have been included in the Supplementary and commented in the main text, citing the corresponding reference. In the main text, we write:

Single-domain configurations with QVH edge states have been considered for single particles in the past \cite{Yao2009}. While the edge states provide a similar dependence $E_v(k_0)$ for vortices (giving rise to associated velocity along the edge), such configuration does not exhibit the same localization potential in transverse direction $E_v(x_0)$ as that shown in Fig. 2(c), and the vortex either crosses the interface and disappears or enters the bulk (see Supplementary Note 5, Supplementary Movies 8, 9). Therefore, the abrupt domain wall between two opposite-staggered lattices that we consider is really an optimal configuration for vortices.

Qualitatively, if the sample is surrounded by “barriers” (corresponding to staggered graphene flake in vacuum), the condensate density in these barriers is zero, and thus the vortices can easily disappear by going towards these regions. Such configuration therefore does not allow to see any interesting behaviour. Therefore, there is no robust behaviour if one uses an exact analogy with the reference mentioned by the referee.

However, we have also tried to surround the “graphene flake” by a region with a condensate (but without any lattice potential). This corresponds to a graphene flake surrounded by a conductor, with narrow isolating barriers. Such configuration does not allow the vortex to escape from the flake, but the effective potential which was providing the vortex binding to the interface is modified: the vortex is not bound any more, it detaches from the interface and goes into the bulk of the sample.

This can also be interpreted using the quasiparticle language: if the vortices are considered as emergent particles, **the effective potential for them is determined by the condensate density, with low-density regions corresponding to the potential minima**. In this case, a flake surrounded by vacuum actually does not correspond to an insulator surrounded by a topologically different insulator,

but rather to an insulator surrounded by a conductor, to which the quasiparticles can easily tunnel. Of course, such configuration is not favourable for the observation of the edge states. On the other hand, a flake, surrounded by a condensate corresponds to an insulator in a vacuum for vortices, but this condensate creates such an important extended repulsive potential for vortices that it perturbs the edge states (which exist only in a narrow range of parameters in a single-domain sample) and does not allow the vortices to localize on the edge. **In both cases, edge states for vortices with an energy corresponding to the bulk gap (for vortices) do not exist any more.**

Therefore, the best configuration is the one already presented in the main text, with a larger interface, providing optimal protection against localized defects.

In the Supplementary, we have added a new figure (showing the spatial energy profile for the vortex) and the following text:

As discussed in the main text, a single-domain sample can also possess QVH edge states with a non-zero group velocity. While this configuration leads to a non-zero velocity of vortices along the interface (inherited from the linear states via the vortex energy dependence on the wavevector $E_v(k_0)$, it does not provide the same double-well localization potential for vortices $E_v(x_0)$ as the one in Fig. 2(c) of the main text (see Fig. 4, black solid line). Indeed, if the flake is surrounded by vacuum (zero condensate density), the transverse potential (roughly given by the condensate density profile) exhibits a steady decrease across the interface (red dashed line), which means that a vortex can escape from the flake into the zero-density region and disappear. This is shown in Supplementary Movie 8. If the flake is surrounded by a constant-density condensate, the potential is modified in the opposite way: the vortex is repelled from the interface into the bulk (blue dotted line, see also Supplementary Movie 9). In both cases, edge states for vortices with an energy corresponding to the bulk gap (for vortices) do not exist anymore. The two-domain structure with an abrupt domain wall represents therefore the optimal configuration for the observation of the QSH analog for vortices.

\item The movie `\textsf{qvh_defect_wide_interface.avi}` (also available at [\url{https://www.youtube.com/watch?v=QNtYwVRY2Ok}](https://www.youtube.com/watch?v=QNtYwVRY2Ok)) shows that in a single-domain sample (with an additional potential on the edge, providing QVH edge states) **\emph{surrounded by vacuum}** the vortex is not protected from going into the zero-density region and disappearing. The position of the vortex is marked with a white cross.

\item The movie `\textsf{vortex_zero.avi}` (also available at [\url{https://www.youtube.com/watch?v=Dzqgeret8Rs}](https://www.youtube.com/watch?v=Dzqgeret8Rs)) shows that if a single-domain sample (with an additional potential on the edge, providing QVH edge states) **\emph{surrounded by vacuum}** is surrounded by a constant density condensate, the vortex is repelled from the interface into the bulk.

3) On page 6, the authors address the winding-valley coupling by taking into account only the 3 atoms of the A type of the central hexagon (in supplementary material). When the vortex core size is large compared with lattice constant, will it be necessary to take into account more atoms, e.g., next nearest neighbour atoms? How does the vortex core size affect the coupling quantitatively?

The restrictions in the analysis are due to the wish to obtain analytical results. Indeed, in this case one can carry out the Fourier transform and show explicitly (and analytically) that the core of the vortex corresponds to a given valley. In the case of a larger core, numerical solution is required. This is what is done in Fig. 1.

Qualitatively, the winding-valley coupling is exact for the central hexagon of a sufficiently large vortex (and all our considerations apply only to such vortices, including numerics). If the vortex size becomes

comparable with a single hexagon, two effects occur: 1) the winding-valley coupling is suppressed because the interactions lead to the filling of the lattice sites of the other type; 2) the vortex becomes trapped on a single hexagon, because jumping to a neighbour hexagon starts to cost a lot of energy. We have improved the discussion of this aspect in the manuscript, adding the following text:

We note that if the vortex size becomes comparable with the lattice parameter $\xi \approx a$, the winding-valley coupling becomes reduced: the vortex acquires a fraction of a different valley. At the same time, it becomes localized on the hexagon center and its mobility is reduced.

4) In Fig. 1, does the vortex core lie exactly at an A type lattice site? It will be helpful to show the lattice sites together with the density profile. And why the vortex density profile has no C3 symmetry?

No, the vortex core is approximately in the center of a hexagon, and this is also the situation that we consider in the analytical treatment. The C3 symmetry is broken because the core is only approximately in the center of the hexagon. But even in that case, the numerical simulation shows that the wavefunction is localized at the valleys.

We have followed the suggestion of the referee: we have entirely remade the figure, obtaining a better symmetry by reaching a better relaxed situation when the vortex is more exactly in the center of the hexagon, and showing the lattice in the real space images. We thank the referee for drawing our attention to this figure, which now allows us to show the explicit shape of the lattice used in the simulations, commenting it in the main text (see the changes below).

5) The authors should explicitly explain why and how they filter the vortex core using Gaussian of different size w in Fig. 1a-c.

The filtering is obtained by a simple multiplication of the solution of the Gross-Pitaevskii equation by a Gaussian function. This allows to obtain the information on different contributions. A large Gaussian has a dominant contribution of the particles far from the core, which are mostly close to the Gamma point of the dispersion (that is, at its minimum). A small Gaussian size allow to see only the vortex core, demonstrating that for the central hexagon all particles are located at a given valley of the reciprocal space, confirming the winding-valley coupling. We now explain it better in the text:

The results of these calculations are shown in Fig. 1, where the black lines show the contours of the pillars corresponding to the potential $U(r)$ (panels (a)-(c)), and the white dashed line shows the 1st Brillouin zone (panels (d)-(e)). To get the information on the vortex core, we apply spatial filtering using a Gaussian of size w .

6) In the last paragraph of page 9, what does "kappa" mean in the second line? It seems that there is no definition given in the main text.

Actually, kappa has been defined in the previous page. It is the width of the interface states.

7) It would be helpful to have some discussions on the finite temperature effect.

This is an interesting and important question. For the particular realization that we consider (exciton-polaritons in a GaAs microcavity at 5 K), the coupling to thermal excitations (phonons) is relatively weak, because of the steepness of the polariton dispersion. Of course, already for the mere observation of polaritons one needs to have the temperature much smaller than the light-matter coupling and the exciton binding energy (both of the order of ~ 10 meV). This is verified by the typical experimental conditions. The main sources of broadening are the disorder and the finite lifetime. The temperature-induced broadening can therefore be neglected.

Thermal and quantum fluctuations are important for the existence of an extended condensate. However, the estimates for polariton condensates show that because of the very small polariton mass these effects should become important only at scales much larger than those discussed in the present work.

We now comment on this in the text:

The mean-field approximation we are using neglects quantum and thermal fluctuations, which reduce the coherence length of the condensate. For the particular system of exciton-polaritons in GaAs cavities at 5 K, the effect of thermal fluctuations is reduced because of the strong decoupling from the phonon reservoir \cite{Wertz2010}. The quantum fluctuations give a theoretical limit for the coherence length of the order of $1\sim\text{mm}$ \cite{Solnyshkov2014}, which makes the mean-field GPE a good approximation at the scale of the lattice we consider ($100\sim\mu\text{m}$). The main sources of broadening, limiting the possibilities of experimental observation, are therefore the disorder and the finite lifetime.

Reviewers' comments:

Reviewer #1 (Remarks to the Author):

The authors have answered all my questions well except one. I do recommend the publication of this work in Nature Commun., but the remaining question needs to be fully addressed in the manuscript before the publication.

- The authors need to explain whether any inter-valley coupling can exist in the current case, although there is valley-vorticity locking. It would be hard to believe that nothing can actually couple vortices and anti-vortices or K and -K. (It is fine to have such mechanisms, and in a real system it is a matter of how to minimize it.)

- The essence of Z2 quantum spin Hall effect is the requirement of time-reversal symmetry instead of spin quantization. Namely, spin does not need to be a good quantum number. In the current case, it is evidently that valley or vorticity needs to be well defined (i.e. uncoupled K/K' or vortex/anti-vortex). Thus, the current work is QVH effect analog instead of QSH effect analog. In addition, the authors have used spin 18 times but valley 46 times and QVH 8 times in the manuscript.

-Therefore, the title should be changed to "QVH effect analog ..." to reflect the actual physics in this paper.

Reviewer #2 (Remarks to the Author):

The authors have satisfactorily addressed all my comments. The presentation has been improved in the revised manuscript. I recommend publication of the manuscript in its present form.

In this document, we provide a point-by-point response to the comment of the referee 1. The changes to the manuscript are marked in blue.

Reviewer #1 (Remarks to the Author):

Referee writes: The authors have answered all my questions well except one. I do recommend the publication of this work in Nature Commun., but the remaining question needs to be fully addressed in the manuscript before the publication.

Authors reply: We thank the referee for the positive recommendation. We have fully addressed the remaining question by extending our discussion and changing the title as suggested by the referee (see below).

- The authors need to explain whether any inter-valley coupling can exist in the current case, although there is valley-vorticity locking. It would be hard to believe that nothing can actually couple vortices and anti-vortices or K and -K. (It is fine to have such mechanisms, and in a real system it is a matter of how to minimize it.)

This is indeed an important question. One of such mechanisms, whose dominant role has been confirmed by the simulations, is already seriously discussed in the text: it is the vortex tunnelling across the interface. Another possible mechanism could be the “locking fault”: if the gap is almost (or completely) closed, the valleys are still well defined, but the valley-winding locking is lost. Both mechanisms are controlled by the size of the gap, which should exceed the broadening from all sources, as already stated in the manuscript.

As to the explicit vortex-antivortex coupling, it is strongly suppressed, because, similar to the quantum-mechanical tunnelling of macroscopic objects, it requires a simultaneous transition to take place for N particles, giving a factor $\exp(-N)$, where $N=10^3 \dots 10^5$. This mechanism is therefore really negligible with respect to the others that we discuss.

We now discuss the “locking fault” and the vortex-antivortex coupling in the manuscript. We added the following phrases at the end of the Discussion section:

If the broadening becomes so strong that it closes the gap, the winding-valley coupling is suppressed. This can be considered as a valley scattering mechanism for a vortex, corresponding to the same restriction for chiral vortex propagation as already discussed above: $\sigma < \Delta$. On the other hand, the winding inversion for the vortex is suppressed even stronger than its tunnelling across the barrier, that is, by a factor $\exp(-N)$, where $N \sim 10^4$ is the total number of particles in the condensate. Such mechanism is therefore completely improbable and does not add any restrictions to the experimental conditions.

We have also modified a phrase in the end of the Results section, to attract the reader’s attention to the different possible situations:

The vortex can be localized on either side of the interface, the latter acting as a barrier preventing the vortex from changing domain and valley (the only valley scattering mechanism remaining for vortices unless the winding-valley coupling is suppressed).

- The essence of Z₂ quantum spin Hall effect is the requirement of time-reversal symmetry instead of spin quantization. Namely, spin does not need to be a good quantum number. In the current case, it is evidently that valley or vorticity needs to be well defined (i.e. uncoupled K/K' or vortex/anti-vortex). Thus, the current work is QVH effect analog instead of QSH effect analog. In addition, the authors have used spin 18 times but valley 46 times and QVH 8 times in the manuscript.

-Therefore, the title should be changed to "QVH effect analog ..." to reflect the actual physics in this paper.

We have followed the recommendation of the referee and changed the title accordingly. The new title reads:

Robust quantum valley Hall effect for vortices in an interacting bosonic quantum fluid

We replaced "spin" by "valley" and removed the word "analog", because the QVH effect at the basis of the work is not an analog, but a genuine QVH effect for photons in a photonic lattice. We have added the word "robust" to stress the contribution of vortices via the winding-valley coupling to the robustness of the effect.